# The Italian Validation of the Beck Cognitive Insight Scale: Underlying Factor Structure in Psychotic Patients and the General Population

**DOI:** 10.3390/ijerph20176634

**Published:** 2023-08-24

**Authors:** Maria Donata Orfei, Desirée Estela Porcari, Gianfranco Spalletta, Francesca Assogna, Fabrizio Piras, Nerisa Banaj, Emiliano Ricciardi

**Affiliations:** 1Molecular Mind Laboratory (MoMiLab), IMT School for Advanced Studies Lucca, Piazza S. Francesco, 19, 55100 Lucca, Italy; desiree.porcari@imtlucca.it (D.E.P.); emiliano.ricciardi@imtlucca.it (E.R.); 2Neuropsychiatry Laboratory, Department of Clinical and Behavioral Neurology, IRCCS Santa Lucia Foundation, Via Ardeatina, 306, 00179 Rome, Italyf.assogna@hsantalucia.it (F.A.); f.piras@hsantalucia.it (F.P.); n.banaj@hsantalucia.it (N.B.); 3Menninger Department of Psychiatry and Behavioral Sciences, Baylor College of Medicine, 1977 Butler Blvd., Houston, TX 77030, USA

**Keywords:** cognitive insight, metacognition, psychometric properties, validation, schizophrenia, general population

## Abstract

Cognitive insight refers to the ability to question one’s judgments and cognitive biases and is underpinned by specific metacognitive processes. The Beck Cognitive Insight Scale was developed to assess cognitive insight and includes two subscales, Self-Reflectiveness and Self-Certainty (SC). The present study aimed to investigate the underlying factor structure of the Italian version of the BCIS in patients with schizophrenia (SZ) and in the general population (GP) for the first time. A cross-sectional design was adopted and a GP sample of 624 subjects and an SZ sample of 130 patients were enrolled. In the SZ group, a two-factor solution was supported. The internal reliability of each factor was satisfactory. Two items were eliminated and one item moved from the SC to the SR subscale. In the GP group, a two-factor solution was highlighted. The internal reliability of each factor was satisfactory. However, four items of the SR subscale were deleted. The Italian-validated version of the BCIS shows different structures for the SZ and the GP and is characterized by different features concerning previous studies. This evidence suggests new interpretations of metacognitive processes in the two populations and implies specific therapeutic approaches.

## 1. Introduction

The concept of cognitive insight [1] was introduced concerning patients with schizophrenia and indicated the ability to monitor, question, and correct one’s erroneous convictions, distorted beliefs, and delusional thinking. In this perspective, cognitive insight is originally strictly related to the insight of illness.

To assess the construct of cognitive insight, Beck and colleagues [1] developed the Beck Cognitive Insight Scale (BCIS), which investigates two sub-dimensions, Self-Reflectiveness (SR) and Self-Certainty (SC). The former concerns the subject’s capacity to appraise cognitive biases, consider alternative explanations, and trust in external feedback; the latter concerns an overconfidence in the validity of one’s beliefs and the unwillingness to modify and correct one’s own opinions and interpretations.

Eventually, the construct of cognitive insight showed to be independent of, although representing a prerequisite of, insight into illness [2,3]. Nowadays, it concerns autonoetic consciousness or the capacity to mentally represent, and consequently become aware of, our continuous subjective life experience as related to specific memories [4,5]. Autonoetic consciousness is strictly underpinned by metacognitive function—that is, the knowledge and reflective capacities an individual has concerning their cognitive emotional and behavioral processes, or, more generally, the ability to think about thinking [6,7,8,9].

This broader interpretation of cognitive insight not only raised a remarkable interest and widespread use of the BCIS for clinical studies, but also showed its potential in the non-clinical general population (GP) [3,10,11,12].

### Validation and Translations of the BCIS

The BCIS has been translated into several languages and administered in different cultural contexts, including European [13,14,15,16], Asian [17,18,19,20,21], and American [22,23] countries.

Furthermore, the BCIS has been validated in several different psychiatric conditions, such as schizophrenia, schizoaffective disorder, major depression with or without psychotic symptoms, bipolar disorder, anxiety disorders, and substance abuse [1,13,14,24,25,26], but also in non-psychiatric medical disorders [27,28]. Finally, an attempt to fix cut-off scores was carried out for clinical purposes [29].

The BCIS has also been widely administered to healthy subjects. So far, a similar factor structure of the BCIS in the GP and in psychiatric patients was found, and the two-factor structure has been confirmed in both populations [23,30].

Despite this, a discrete variability in reliability results, statistical procedures, and pragmatical considerations emerges in the literature [31]. A point deserving attention is a controversy about the appropriateness of some items, which in some cases led to the depletion of some statements in both the psychiatric population and the healthy subject validation procedures [13,32,33]. 

Orfei and colleagues [16] published an Italian translation of the scale, validated by both a back translation with an English mother-tongue expert in the field and an evaluation of the items by two judges regarding semantic congruence, content validity, and theoretical relevance. Assessment of interrater reliability for raters was in the excellent to good range for all the items, with intraclass correlations ranging from 0.80 to 0.93.

However, the psychometric properties of this version of the BCIS have yet to be tested. The main aim of this study was to investigate the underlying factor structure of the Italian version of the BCIS both in patients with schizophrenia (SZ) and in the GP.

## 2. Materials and Methods

### 2.1. Study Design and Setting

A cross-sectional design was adopted. All participants were provided with a detailed description of the experimental procedures and required consent before participating in the study. Participation was anonymous, voluntary, and not rewarded in any form. Each subject could fill out the questionnaire only once. The study was conducted in accordance with the Declaration of Helsinki and under research protocols approved by local Ethical Committees (Scuola Normale Superiore and Scuola Superiore Sant’Anna Joint Ethical Committee: Protocol No. 04/2021).

The General Population sample was recruited by two different modalities: partly by an online survey and partly by in-person recruitment. The online survey was carried out in the months of November and December 2021. The in-person recruitment of healthy volunteers was carried out at the Neuropsychiatry Lab of the IRCCS Santa Lucia Foundation from November 2021 to March 2022. The sample of patients with a diagnosis of schizophrenia was fully recruited from June 2021 to December 2022 at the Neuropsychiatry Lab of the IRCCS Santa Lucia Foundation (see below for further details).

### 2.2. Participants

#### 2.2.1. General Population (GP)

For all the GP groups, the research participation was voluntary and the inclusion criteria were: (a) age greater than or equal to 18 years, (b) Italian mother tongue or high-level knowledge of the Italian language, (c) at least 8 years of education.

For the online survey, an initial sample of 2367 employees of a large Italian banking group was invited. The survey was anonymous, as each participant was assigned an alphanumeric code to ensure the confidentiality of information. Questionnaires were evenly distributed across the national territory. Given the online distribution, no anamnestic information about mental health was collected for these subjects.

Regarding the in-person recruitment, 150 healthy volunteers were recruited at the Neuropsychiatry Lab of the IRCCS Santa Lucia of Rome. These subjects were assessed by one senior clinical psychiatrist. For the in-person GP sample, for whom the collection of anamnestic data was possible, additional exclusion criteria were past or present major medical or neurological illnesses, psychiatric disorders, or mental retardation [34,35].

#### 2.2.2. Patients with Schizophrenia (SZ)

One hundred and fifty consecutive SZ outpatients according to DSM-VI [34,35] were preliminarily recruited at the IRCCS S. Lucia of Rome. All patients were diagnosed by one senior clinical psychiatrist. Inclusion criteria were: (1) age between 18 and 65 years, (2) at least 8 years of education, (3) no dementia or cognitive deterioration according to the DSM V- criteria or Mini-Mental State Examination (MMSE) [36] scores lower than 24, which is consistent with normative data in the Italian population [37]. This criterion was selected to exclude patients with cognitive deterioration, which could represent a confounding variable in the interpretation of results. Exclusion criteria were: history of alcohol or drug dependence or traumatic head injury, any past or present major medical or neurological illness, any additional psychiatric disorder, mental retardation, or dementia.

The Positive and Negative Syndrome Scale (PANSS) [38] was administered by a senior clinical psychiatrist (G.S.) to rate the severity of psychopathological symptoms. PANSS ratings were obtained considering the last week’s period from the assessment.

Extrapyramidal side effects due to current treatment were assessed by the Simpson-Angus Rating Scale (SARS) [39], and the Abnormal Involuntary Movements Scale [40] was administered to assess the presence of tardive dyskinesias. No patient suffered from tardive dyskinesias.

All patients were receiving stable oral doses of one or more atypical antipsychotic drugs such as risperidone, quetiapine, and olanzapine. Antipsychotic dosages were converted to estimated equivalent dosages of olanzapine by use of a standard table [41]. All patients were in their phase of stable clinical compensation.

### 2.3. Beck Cognitive Insight Scale (BCIS)

The BCIS is a standardized, self-rated instrument composed of 15 items that rely on two sub-factors, SR and SC. SR expresses the willingness to acknowledge fallibility, self-appraise, and consider external feedback, while SC expresses firm confidence in one’s judgments and opinions. A subscale of nine items assesses SR (e.g., “If somebody points out that my beliefs are wrong, I am willing to consider it”), while a subscale of six items assesses SC (e.g., “I can trust my judgment at all times”). A third index, R-C, resulting from the SR score minus the SC score, reflects the balance between the two components.

### 2.4. Statistical Analyses

All analyses were performed using IBM SPSS v.27. Descriptive statistics were conducted on sociodemographic variables, with the chi-squared test for the nominal variable (gender) and independent sample *t*-test for the continuous variable (age). An independent sample *t*-test was also performed to compare SZ and GP on BCIS scores. The significance of these comparisons was set at *p* < 0.05.

The Cronbach’s alpha, the McDonald’s omega tests, and the inter-item correlation were performed to test questionnaire reliability. Keiser-Meyer-Olkin (KMO) and Bartlett’s sphericity tests were used to evaluate the adequacy and suitability of the samples (SZ and GP) before performing the factor analysis. A principal components analysis (PCA) was performed to extract factors with eigenvalues greater than or equal to 1.0. To clarify the interpretation, exploratory factor analyses (EFA) with orthogonal rotation (varimax) were performed; the acceptable factor loading index was set at ≥|0.40|.

## 3. Results

From the initial online panel, 487 subjects accepted to participate in the study and completed the online survey; from the initial sample who underwent the in-person psychiatric screening at the IRCCS Santa Lucia Foundation, three subjects were excluded because of a dementia diagnosis or an MMSE score lower than 27, four were excluded because they were diagnosed with comorbid substance use disorders (including the use of cannabis), five were excluded because of a major medical illness or a neuropsychiatric disorder, and one was excluded because of previous traumatic brain injury. Thus, 137 subjects were considered eligible. Summing up the two GP groups, the actual final GP sample was composed of 624 subjects.

About the SZ group, from the initial sample, 11 were excluded because they were diagnosed with comorbid substance use disorders, 5 were excluded because they were cognitively deteriorated, 3 were excluded because of comorbid medical or neurological illness, and 1 was excluded because of previous traumatic brain injury. Thus, 130 consecutive outpatients were included in the final SZ sample.

The size of both samples is adequate to support EFA results [42,43,44].

Descriptive statistics of sociodemographic variables (age and gender), BCIS indexes, and clinical data (PANSS scores and illness duration) are shown in Table 1.

### 3.1. Reliability of the BCIS in the SZ and the GP Groups

The Italian translation of the BCIS, in its original formulation, showed slightly acceptable internal consistency in SZ, both in the SR (Cronbach’s *alpha* = 0.583, McDonald’s *omega* = 0.587) and SC subscales (Cronbach’s *alpha* = 0.608, McDonald’s *omega* = 0.616). In the GP group, slightly acceptable reliability values were found for SR (Cronbach’s *alpha* = 0.564, McDonald’s *omega* = 0.566) and SC (Cronbach’s *alpha* = 0.628, McDonald’s *omega* = 0.630) (Appendix A). Furthermore, no threshold inter-item correlations (≥0.800) were found in SZ and GP (Appendix A).

### 3.2. PCA and EFA in the SZ Group

The KMO (0.609) and Bartlett’s sphericity test results (χ^2^ = 268.656 *p* = 0.000) showed that the data were suitable for factor analysis. The PCA showed six eigenvalues greater than or equal to 1.0, accounting for 62% of the total variance explained (Figure 1).

To explore the factorial structure of the BCIS, all 15 items of the instrument were subjected to EFA with orthogonal rotation (varimax).

The first solution according to the scree plot showed 12 items and six factors, where two factors (4 and 6) were composed of 1 item only. The internal consistency was calculated for factors with several items greater than or equal to 2: factor 1 (2 items; Cronbach’s *alpha* = 0.521, McDonald’s *omega* = 0.521), factor 2 (3 items; Cronbach’s *alpha* = 0.534, McDonald’s *omega* = 0.570), factor 3 (3 items, Cronbach’s *alpha* = 0.500, McDonald’s *omega* = 0.511), and factor 5 (2 items; Cronbach’s *alpha* = 0.470, McDonald’s *omega* = 0.470) (Appendix A). Due to this questionable factor solution, the EFAs with four, three, and two forced factors were conducted (Appendix A, Appendix A).

The four forced factors solution showed 12 items and the following internal consistency indexes for each factor: factor 1 (3 items; Cronbach’s *alpha* = 0.573, McDonald’s *omega* = 0.534), factor 2 (4 items; Cronbach’s *alpha* = 0.521, McDonald’s *omega* = 0.506), factor 3 (3 items; Cronbach’s *alpha* = 0.455, McDonald’s *omega* = 0.444), and factor 4 (2 items; Cronbach’s *alpha* = 0.000, McDonald’s *omega* = −0.373).

Then EFA with three forced factors solution was performed reporting 13 items distributed in three factors with the following internal consistency indexes: factor 1 (5 items; Cronbach’s *alpha* = 0.624, McDonald’s *omega* = 0.618), factor 2 (4 items; Cronbach’s *alpha* = 0.521, McDonald’s *omega* = 0.506), and factor 3 (4 items; Cronbach’s *alpha* = 0.509, McDonald’s *omega* = 0.487).

Lastly, the EFA with two forced factors showed 13 items and the following reliability indexes: factor 1 (8 items; Cronbach’s *alpha* = 0.610, McDonald’s *omega* = 0.607) and factor 2 (5 items; Cronbach’s *alpha* = 0.624, McDonald’s *omega* = 0.618) (Appendix A). The two factors were respectively renamed: *Self-unreliability*, i.e., the metacognitive belief of poor trust in the self; *Self-certainty*, consisting of part of the same items of the SC subscale of Beck’s original version (Figure 2).

### 3.3. PCA and EFA in the GP Group

The KMO (0.717) and Bartlett’s sphericity test results (χ^2^ = 1331.186 *p* = 0.000) showed the adequacy of the GP sample. The PCA showed four eigenvalues greater than or equal to 1.0, accounting for 50% of the total variance explained (Figure 3).

To explore the factorial structure of the BCIS in the GP sample, all 15 items of the instrument were subjected to EFA with orthogonal rotation (varimax).

The first solution according to the scree plot showed 14 items and four factors, whereas the fourth factor was composed of only one item. The internal consistency was calculated for factors with several items greater than or equal to 2: factor 1 (5 items; Cronbach’s *alpha* = 0.500, McDonald’s *omega* = 0.414), factor 2 (5 items; Cronbach’s *alpha* = 0.641, McDonald’s *omega* = 0.636), and factor 3 (3 items, Cronbach’s *alpha* = 0.526, McDonald’s *omega* = 0.511) (Appendix A). Given this questionable factor structure, an EFA with three and two forced factors was performed (Appendix A).

The three forced factors solution showed 14 items and the following factors internal consistency: factor 1 (5 items; Cronbach’s *alpha* = 0.666, McDonald’s *omega* = 0.667), factor 2 (6 items; Cronbach’s *alpha* = 0.630, McDonald’s *omega* = 0.628), and factor 3 (3 items; Cronbach’s *alpha* = 0.526, McDonald’s *omega* = 0.511).

The EFA with two forced factors obtained 10 items and the reliability of the following factors: factor 1 (5 items; Cronbach’s *alpha* = 0.666, McDonald’s *omega* = 0.667) and factor 2 (5 items; Cronbach’s *alpha* = 0.641, McDonald’s *omega* = 0.636). Factors 1 and 2 were respectively named: *Self-doubt*, i.e., a critical inspection of own beliefs; *Self-certainty*, overlapping the SC subscale of Beck’s original version (Figure 4).

## 4. Discussion

The main aim of the study was to examine the underlying factor structure of the Italian version of the BCIS both in SZ patients and the GP. We obtained two main results: (a) in the SZ group, the analyses supported the efficiency of a two-factor model, which was similar—although not fully comparable—to Beck’s original structure of the scale; and (b) in the GP group, a two-factor model best fitted the data, although it implied a remarkably different structure concerning Beck’s original version.

### 4.1. Results in the SZ Group

In the SZ group, the two-factor solution turned out to be the best fitting for the data because it was supported by an acceptable internal reliability level, and the data were consistent with previous evidence.

Despite the exclusion of two items from the SR subscale, our two-factor model mostly resembled the original Beck description of the questionnaire. To note, the EFA showed that the first factor included seven original SR items along with an item drawn from the SC subscale (“I cannot trust other people’s opinion about my experiences”). This statement originally aimed to elicit an attitude of mistrust in other people. This apparent incongruence might cast light on the peculiar metacognitive process explored by the SR subscale in our model. In fact, in SZ patients, it might mirror a deep feeling of unreliability of one’s judgment, to the extent that even one’s understanding of other peoples’ opinions is doubtful. Thus, in SZ patients, high scores on this factor would indicate a radical mistrust in one’s judgment ability, rather than a healthy openness to external feedback. In this perspective, this dimension would be better renamed as “Self-Unreliability”. This would be consistent with several previous studies that highlighted that in psychotic patients, SR may be healthy to a point, after which high levels may be associated with depressive thinking and self-stigma [45,46,47,48,49] and even jeopardize the quality of life [46,50,51].

Indeed, our study not only provided results partially different from previous evidence in terms of internal reliability [31] and data interpretation, it also showed that the reliability values—as defined by Cronbach’s alpha or McDonald’s omega—were not very high, although they were still fairly acceptable. However, we would like to bring attention to some data. Regarding the SZ sample, the alpha values of our final 2-factor structure were 0.610 and 0.624, respectively. Indeed, previous studies showed a fair variability in results, and any case obtained not much higher values. Beck and colleagues [1] reported an alpha of 0.68 for SR and 0.60 for SC; Favrod [14] described a French translation with an alpha of 0.73 (SR) and 0.62 (SC); Greenberger [2] obtained an SR characterized by alpha = 0.61 and 0.84, respectively. Among confirmatory analyses, Saguem [19] provided reliability scores of 0.60 for Sr and 0.53 for SC, and Pedrelli [25] of 0.70 and 0.55, respectively. This difference in results is attributable to the features of the samples recruited, namely psychiatric diagnosis and duration of illness. Most of the studies enrolled psychiatric patients with different diagnoses, ranging from schizophrenia to schizoaffective, bipolar, and major depressive disorders, with or without psychotic features, and even not otherwise specified psychotic features [1,13,14,17,47]. Conversely, we selected a sample that was highly homogeneous in terms of diagnosis, clinical picture, and medical treatment in order to not to blur results. Interestingly, Gutierrez and colleagues [15], different from the aforementioned studies and similar to ours, included only patients with a diagnosis of schizophrenia and reported alpha values of 0.59 for SR and 0.62 for SC and had to delete three items from the original version; thus, they showed evidence consistent with our results.

Finally, our two-factor structure required the depletion of two items, which was a not entirely new solution. Some items of the original Beck’s version have been previously debated. For instance, Pedrelli and colleagues [25], in their study on patients with schizophrenia and schizoaffective disorder, slightly modified the phrasing of an item from the original version, which referred to “unusual experiences” as “unpleasant experiences”. The suitability of the original version of this item for the GP is also questioned by Martin and colleagues [32]. Furthermore, in Pedrelli’s study, although item 14 is lower than the cutoff loading of 0.30 used for determining whether an item contributed significantly to a specific factor, it was nonetheless saved.

### 4.2. Results in the GP Group

In the GP group, the forced two-factor solution was characterized by the good internal reliability level of each factor. However, this higher consistency was obtained by the elimination of five SR items, which significantly altered the original structure of the scale. Notably, most of the SR factors deleted by the EFA concerned external feedback. We might speculate that the cognitive insight in the GP is underpinned by a metacognitive process that is drastically different from that of the SZ. In fact, in SZ patients, external feedback, up to a certain point, can represent a benchmark due to the untrustworthiness of one’s judgment function. Conversely, GP self-reflection would mostly pertain to a critical attitude toward one’s judgment biases, similar to the metacognitive construct of private self-consciousness or the inspection and evaluation of our thinking and feelings [51,52]. Different neuropsychological correlations of SR in the SZ and the GP highlighted in previous studies [17,53,54,55] would support this different role of self-reflection in the two populations.

This is the first validation study pointing out this structure of the SR in the GP. The few previous studies suggested a depletion of items, and were merely focused on the sentences implicitly concerning psychotic experiences [13,25,32], which was deemed inappropriate for healthy subjects. Engh [13] raised doubts about the suitability of items 3, 5, 6, and 15 because in his control group, they were left out to a large degree, possibly because referring to psychotic experiences was difficult to answer for this group. Thus, he concluded that in the case of comparison between controls and other clinical samples, the four items referring to psychotic experiences should not be used. Uchida [20] decided to keep item 15, although it did not reach the loading threshold, in order to guarantee consistency with Beck’s original results. One of the most different interpretations of the BCIS, when applied to healthy subjects, is described by Kao and colleagues [17]. The authors’ factorial analysis pointed out that two items originally of the SC subscale (items 10 and 11) were now included in the SR subscale, and that two items originally about the SR subscale were now better loaded on the SC subscale. Moreover, the factorial structure is quite different from Beck’s description, to the extent that the authors suggest renaming SR and SC as “reflective attitude” and “certain attitude”, respectively. Thus, although their principal components analysis supported a two-factor solution, the pattern of intercorrelation suggested that cognitive insight comprises two or even more related yet partially independent components.

Our evidence can be traced back to the characteristics of the samples enrolled because our GP group was, on average, older and better balanced for gender than in other studies [23,32]. In particular, older subjects can benefit from greater self-experience and consequently be more aware of one’s judgment biases; thus, they are less concerned with external feedback. This is not to say that the different statistical approaches adopted may play a role, as some authors performed a confirmatory factor analysis rather than a validation procedure [32].

As for the SZ group, for the GP group, we also obtained reliability indices that are of moderate magnitude at first glance. Our final two-factor structure was characterized by alpha scores of 0.666 for the first factor (renamed as Self-doubt) and 0.641 for the second factor (SC). As debated for the SZ group, our values are not very unlike those reported in previous studies. Warman [30] showed that, in a population of undergraduate students, the alpha values of the BCIS were 0.62 for SR and 0.61 for SC. Engh [13], in a much smaller sample of subjects that were supposed to not have psychiatric disorders, described the Norwegian translation of the BCIS as characterized by 0.73 (SR) and 0.63 (SC). A wider healthy sample was enrolled by Buchy and colleagues [23] to validate the BCIS in the Canadian community, and alpha scores of 0.68 for SR and 0.65 for SC emerged. Other studies reported higher reliability indices. Among others, Kao and colleagues first reported Cronbach’s values of 0.70 (SR) and 0.69 (SC) [17] and later of 0.75 and 0.78, respectively. Also, Martin [32] highlighted alpha values of 0.74 for SR and 0.75 for SC in a sample of undergraduate students. However, we cannot help but note that most of the previous studies are underpinned by very young subjects, and by samples frequently unbalanced for gender. This evidence allows us to suppose that our data are fairly consistent with the literature and underpinned by a reliable sampling approach.

### 4.3. Limitations

Some limitations must be considered when interpreting the results of the present study. First, our main concern is selection bias. As the majority of our GP group was constituted by employees of a single Italian banking group, the results may not represent the general Italian population. Moreover, the difference between the figures of the online panel and those of the actual sample may seem impressive at first glance. To note, those who have accepted our invitation to participate in the study as volunteers may be more sensitive toward psychological issues. However, some points have to be considered. The participation was completely voluntary and anonymous. Consequently, no data are available about the characteristics of the panel invited and much less about those of the subjects who rejected to get involved in the online survey. To address these potential sources of bias, we also included an additional GP group external from the bank to increase the representativeness of results and to cautiously speculate about the possibility of extending our results to wider segments of the Italian population.

Second, the premorbid cognitive and psychiatric status was not documented for the online-recruited GP subsample; thus, we cannot exclude the presence of psychological discomfort symptoms. This is why we opted for the “GP” denomination of the group, rather than “healthy participants”. However, the inclusion of volunteers who underwent a psychiatric screening in the GP group, which excluded any present or past psychiatric issues, is expected to counterbalance the missing information on the other part of the sample.

Third, on one side, the strict inclusion criteria defined for the SZ group ensured the validity of our results; on the other side, it limited their generalizability to other psychotic disorders. Future studies would cast light on this point, enrolling patients with different diagnoses.

Fourth, our study aimed to specifically explore the underlying factor structure of the Italian version of the BCIS, but did not include the investigation of additional psychometric properties, such as test–retest stability and discriminant validity. Further studies to deepen these psychometric properties are required to provide a fully exhaustive perspective on the issue.

## 5. Conclusions

The BCIS is an efficient tool to assess general metacognitive processes underpinning autonoetic consciousness in diverse psychiatric disorders as well as non-clinical subjects [17].

The results presented here provide a benchmark for cognitive insight in Italian GP and SZ patients, allowing us to speculate on different metacognitive processes. Notably, treatments for patients with psychosis, and in particular the Metacognitive Training [56], frequently aim to improve cognitive insight because this seems to affect overall treatment outcomes and recovery [10,57]. The dimension that we named Self-Unreliability may orient the therapeutic approach for SZ patients to improve insight into illness, supporting patients to consider external feedback as sources of information to integrate into one’s cognitive schemata, consequently reinforcing confidence in managing information and evidence. On the other hand, in the GP, the management of autonoetic knowledge highlighted in our study may suggest a psychotherapeutic approach toward encouraging to take external feedback into greater consideration, as well as a realistic self-evaluation.

Nonetheless, the evidence highlighted in our study poses a pragmatic problem for research. We may wonder whether two different forms of the same questionnaire can be administered in a unique study to assess the same construct in GP and SZ patients. In particular, the depletion of some items from the original form of the scale cannot be disregarded. We may wonder what factors led to this evidence. It might be traced back to the translation into the Italian language, but we are not inclined to consider this hypothesis because the translation process was supported by good results. Notably, we recommend to not interpret the evidence that emerged from our study as questioning the purposefulness of the BCIS per se. Rather, the partly controversial debate of the underlying factor structure of the scale, with the contribution of our additional data, would require further deepening and clarification in future studies. In waiting for further definitive evidence, we suggest administering the classical form of the BCIS.

Interestingly, our study points out that the most unstable dimension of cognitive insight is SR. This is possibly due to the multifactorial nature of this construct.

Different from SR, we confirm the consistency of the SC dimension across different studies, samples, cultural contexts, and statistical approaches, which consequently seems to investigate a more stable construct than SR.

## Figures and Tables

**Figure 1 ijerph-20-06634-f001:**
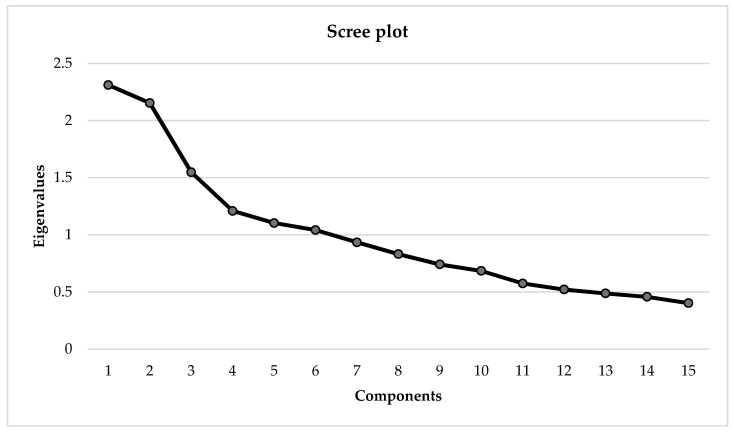
PCA Scree plot with eigenvalues greater than or equal to 1.0 in the SZ group. PCA = Principal components analysis; SZ = Patients with schizophrenia.

**Figure 2 ijerph-20-06634-f002:**
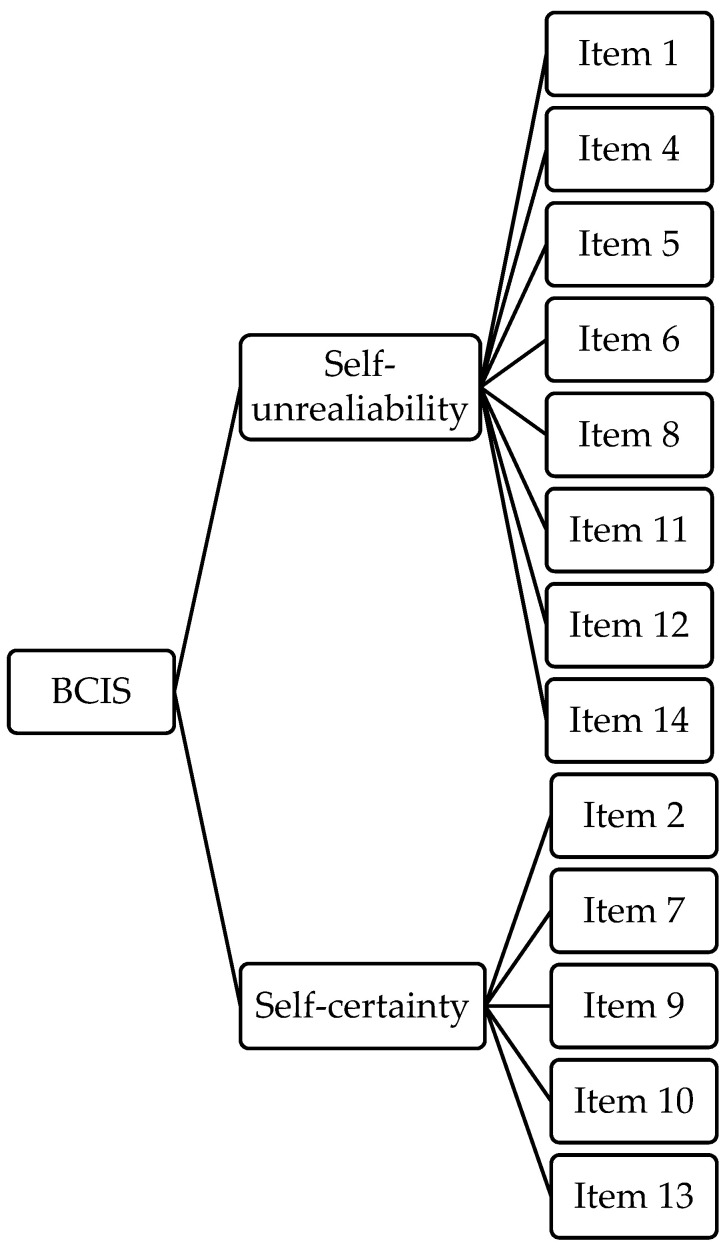
The path diagram of the two-factor model of the BCIS in SZ. Survivor item distribution after the EFA in the SZ group. BCIS = Beck Cognitive Insight Scale; SZ = Patients with schizophrenia.

**Figure 3 ijerph-20-06634-f003:**
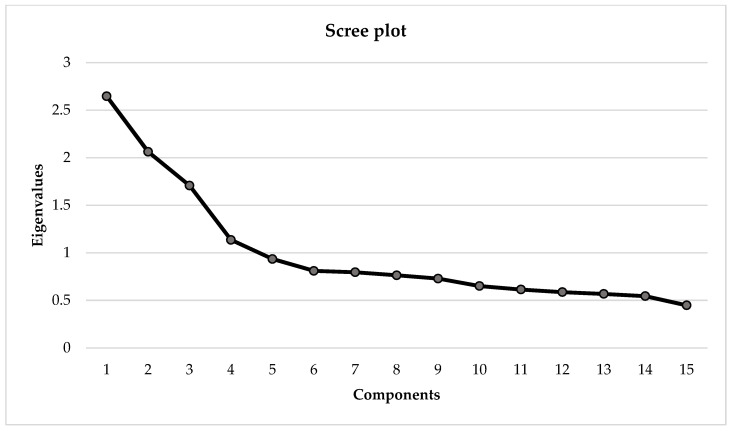
PCA Scree plot with eigenvalues greater than or equal to 1.0 in the GP group. PCA = Principal components analysis; GP = General population.

**Figure 4 ijerph-20-06634-f004:**
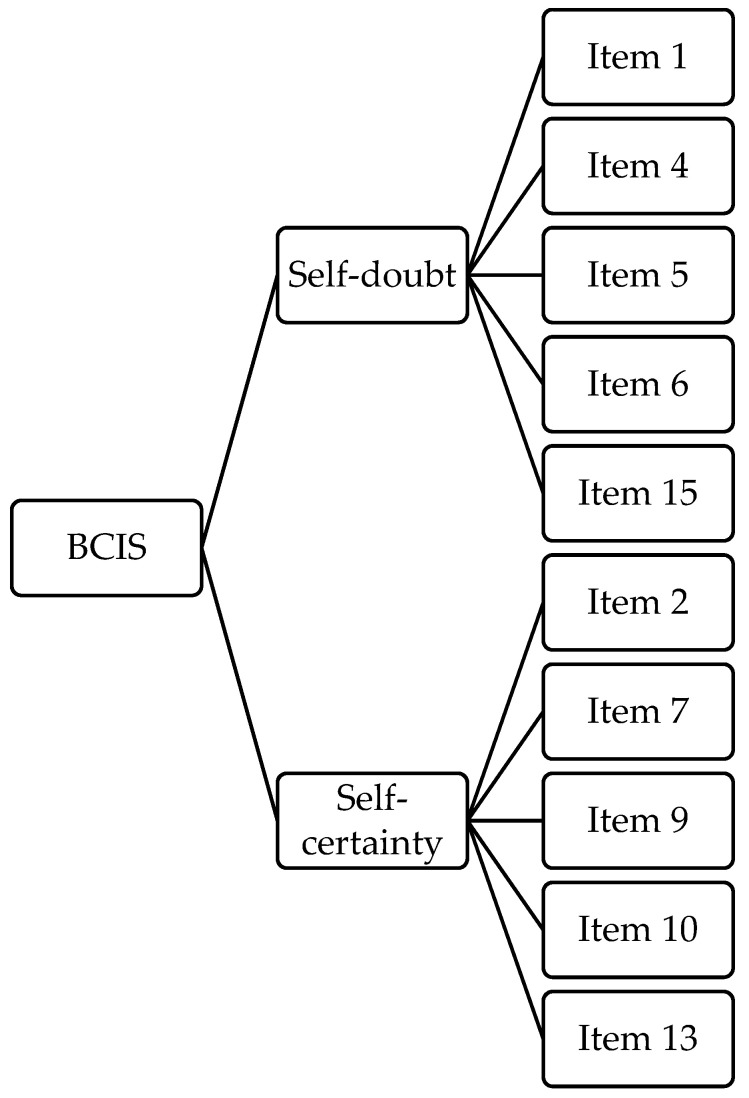
The path diagram of the two-factor model of the BCIS in GP. Survivor item distribution after the EFA in the General Population. BCIS = Beck Cognitive Insight Scale; GP = General population.

**Table 1 ijerph-20-06634-t001:** Descriptive statistics of socio-demographic variables, BCIS indexes, and clinical data.

	GP *N* = 624	SZ *N* = 130	*p*-Value
Age, years (M ± SD)	46.3 ± 9.7	37.9 ± 10.5	<0.001
Female Gender N (%)	261 (42%)	35 (27%)	0.002
BCIS SR (M ± SD) *	10.6 ± 3.2	13.6 ± 4.0	<0.001
Range	Min 0/Max 22	Min 0/Max 27	-
BCIS SC (M ± SD) *	8.1 ± 2.7	8.7 ± 3.7	<0.001
Range	Min 0/Max 14	Min 0/Max 18	
Illness duration in years (M ± SD)	-	14.7 ± 9.9	-
PANSS GPsy (M ± SD)	-	42.1 ± 10.8	-
Range		Min 19/Max 79	
PANSS Pos (M ± SD)	-	19.9 ± 5.9	-
Range		Min 7/Max 34	
PANSS Neg (M ± SD)	-	20.8 ± 7.2	-
Range		Min 7/Max 40	

Significant results (p < 0.05) are highlighted in bold. GP = General population; SZ = Patients with schizophrenia; PANSS = Positive and Negative Syndrome Scale; M = Mean; GPsy = General Psychopathology scale; Pos = Positive symptoms scale; Neg = Negative symptoms scale; SD = Standard deviation; BCIS = Beck Cognitive Insight Scale; SR = Self-Reflectiveness; SC = Self-Certainty. * BCIS scores are calculated according to Beck’s version of BCIS.

## Data Availability

Not applicable.

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
