# Peer review of "The Italian Validation of the Beck Cognitive Insight Scale: Underlying Factor Structure in Psychotic Patients and the General Population"

_ijerph, 2023, doi:10.3390/ijerph20176634_

Round 1

Reviewer 1 Report

Background:

The main aim of the study was to examine the psychometric properties of the Italian version of the Beck Cognitive Insight Scale (BCIS) both in SZ patients and in a sample of the general population. The title of the paper should be modified to state that the aim was to examine the underlying factor structure. Psychometric properties include additionally reliability, test-retest stability, discriminant validity.

Methods:

The GP sample queried was very large, but only 487 subjects were included. This needs to be discussed in terms of the true representation of the general population sample. Can they compare the respondents to the non-respondents on some meaningful variables? It would help to have more table formatted data on the two samples’ BCIS data, such as the range of the total rating and the two subscales’ ratings. They included data on the patient sample on EPS, which is not clear as to what the purpose was for this data. It would have been much more interesting to have quantitative data on the psychopathology of the patient sample. Same for the medication data, which is not used in the study. All of the patients were on oral antipsychotics; did they exclude patients on long acting injectables intentionally?

Results:

The Cronbach alpha were generally not impressive (for both samples) and should be explained in their discussion. They report the two factor structure which was found for both samples. They also should show in a Table the factor loadings for each item for both samples. This would help the reader to get a better idea what items went with what factor. Were there items which loaded on both factors (SR and SC)? The GP sample lost five items from the original subscale. Again, one would want to see what the factor loadings of these items were and which exact items.

Discussion:

They showed that their analysis in the SZ supported a 2-factor model, similar, but not equal to Beck’s original structure of the scale. Could they speculate more whether the Italian language played a role in this difference? In the limitations, they need to indicate that they did not examine reliability, test-test stability and discriminant validity (see above).

A few English edits are needed.

Author Response

We sincerely thank the Reviewer for his comments and suggestions that helped us to improve significantly our paper. Here below our answers to his/her notes:

Background:

 The main aim of the study was to examine the psychometric properties of the Italian version of the Beck Cognitive Insight Scale (BCIS) both in SZ patients and in a sample of the general population. The title of the paper should be modified to state that the aim was to examine the underlying factor structure. Psychometric properties include additionally reliability, test-retest stability, discriminant validity.

We thank the Reviewer for the specification. Consistently, we have slightly changed the wording of the title, in the abstract (line 19), in the Introduction (line 100), and in the Discussion (line 350). In particular, in the Limitation paragraph, we have stated that: “Fourth, our study aimed at exploring specifically the underlying factor structure of the Italian version of the BCIS, but did not include the investigation of additional psychometric properties, such as test-retest stability, and discriminant validity. Further studies to deepen these additional psychometric properties are required to provide a fully exhaustive perspective on the issue.” Indeed, a study of reliability was run in our paper (lines 467-471).

Methods:

The GP sample queried was very large, but only 487 subjects were included. This needs to be discussed in terms of the true representation of the general population sample. Can they compare the respondents to the non-respondents on some meaningful variables?

We do understand the Reviewer’s concern about the online data, since the difference between the figures of the online panel and those of the actual final sample at first glance may seem impressive. However, some points have to be considered. The participation was completely voluntary and anonymous, as stated in the Methods section (lines 107; 126). Consequently, no data are available about the characteristics of the panel invited and much less about those of the subjects who rejected to get involved in the study. To note, our aim was not to validate the BCIS in the population of bank employees, but rather in the Italian general population, this is why we included also an additional GP group external from the bank, to increase the representativeness of results. Nonetheless, we are aware of possible selection bias, as we have stated in the Limitations section. However, the Reviewer’s comment is quite reasonable and the point requires to be disclaimed, so we included these arguments in the Limitations section as follows: “First, our main concern is selection bias. As the majority of our GP group was constituted by employees of a single Italian banking group, the results may not represent the general Italian population. Moreover, the difference between the figures of the online panel and those of the actual sample at first glance may seem impressive. To note, those who have accepted our invitation to participate in the study as volunteers may be more sensitive toward psychological issues. However, some points have to be considered. The participation was completely voluntary and anonymous. Consequently, no data are available about the characteristics of the panel invited and much less about those of the subjects who rejected to get involved in the online survey. To address these potential sources of bias, we included also an additional GP group external from the bank, to increase the representativeness of results and to cautiously speculate about the possibility of extending our results to wider segments of the Italian population” (lines 444-455)

It would help to have more table formatted data on the two samples’ BCIS data, such as the range of the total rating and the two subscales’ ratings.

Consistently with the Reviewer’s suggestion, in Table 1 we reported the two BCIS sub-scales scores (the BCIS does not provide a total score) both for the general population and the patients with schizophrenia, calculated by Beck’s original instructions. Also, to facilitate the understanding of results we have indicated the minimum and maximum scores of the two subscales obtained in our samples.

They included data on the patient sample on EPS, which is not clear as to what the purpose was for this data. It would have been much more interesting to have quantitative data on the psychopathology of the patient sample. Same for the medication data, which is not used in the study.

We thank the Reviewer for having highlighted the missing psychopathological data. We agree that this is remarkable information for the reader to have a more definite picture of the SZ sample as well as to compare our evidence with those of future studies. Consistently, in Table 1 we have included also the PANSS scores (Total, Positive, and Negative symptoms scales), each accompanied by minimum and maximum scores got in the sample. For the sake of clarity, we have provided information about extra-pyramidal effects, dyskinesias, and treatment to provide the reader and the scientific community with all the data necessary to evaluate our sample characteristics. As stressed in the paper, frequently in the literature some missing or ambiguous information about the subjects' characteristics, especially when dealing with psychiatric disorders, does not allow the reader to understand correctly possible discrepancies in results.

All of the patients were on oral antipsychotics; did they exclude patients on long acting injectables intentionally?

No, we did not intentionally exclude patients with therapeutical approaches other than oral. More simply, the patients attending the IRCCS Santa Lucia Foundation clinic were usually treated with oral pharmacological therapies.

Results:

The Cronbach alpha were generally not impressive (for both samples) and should be explained in their discussion.

We agree with the Referee that the reliability values as defined by Cronbach’s alpha or McDonald’s omega, although fairly acceptable, are not very high. However, the accurate analysis of the literature allows us to bring attention to some data supporting the consistency of our results with those of previous studies. We have outlined arguments for this point about the SZ group as follows: “To note, the reliability values, as defined by Cronbach’s alpha or McDonald’s omega, although fairly acceptable, are not very high. However, we would like to bring attention to some data. About the SZ sample, the alpha values of our final 2-factor structure were 0.610 and 0.624, respectively. Indeed, previous studies showed a fair variability in results, and any case obtained not much higher values. Beck and colleagues [1] reported an alpha of 0.68 for SR and 0.60 for SC, Favrod [14] described a French translation with an alpha of 0.73 (SR) and 0.62 (SC), while Greenberger [2] obtained an SR characterized by alpha=0.61 and 0.84, respectively. Among confirmatory analyses, Saguem [19] provided reliability scores of 0.60 for Sr and 0.53 for SC, and Pedrelli [25] of 0.70 and 0.55, respectively. Interestingly, Gutierrez and colleagues [15], who, differently from the above-mentioned studies and similarly to ours, included only patients with a diagnosis of schizophrenia, reported alpha values of 0.59 for SR and 0.62 for SC and had to delete three items from the original version, thus showing evidence consistent with our results. These data not only highlight that alpha reliability indices for the BCIS are not very high in most of the studies, but also support the hypothesis that a fair variability in results depends on the criteria of inclusion and the population considered” (lines 384-399).

On the other hand, about the GP group we have included the following paragraph: “Similarly to what emerged for the SZ group, for the GP group too we obtained reliability indices that at first glance are of moderate magnitude. Our final 2-factor structure was characterized by alpha scores of 0.666 for the first factor (renamed as Self-doubt) and 0.641 for the second factor (SC). As debated for the SZ group, our values are not very unlike those reported in previous studies. Warman [51] showed that, on a population of undergraduate students, alpha values of the BCIS were 0.62 for SR and 0.61 for SC. Engh [13], in a much smaller sample of subjects supposed not to have psychiatric disorders, described the Norwegian translation of the BCIS as characterized by 0.73 (SR) and 0.63 (SC). A wider healthy sample was enrolled by Buchy and colleagues [23] to validate the BCIS in the Canadian community and alpha scores of 0.68 for SR and 0.65 for SC emerged. Other studies reported higher reliability indices. Among others, Kao and colleagues first reported Cronbach’s values of 0.70 (SR) and 0.69 (SC) [17] and later of 0.75 and 0.78, respectively. Also, Martin [50] highlighted alpha values of 0.74 for SR and 0.75 for SC in a sample of undergraduate students. However, we cannot help but note that most of the previous studies are underpinned by even very young subjects, and by samples frequently unbalanced for gender. This evidence allows us to suppose that our data are fairly consistent with the literature and underpinned by a reliable sampling approach” (lines 424-440).

They report the two factor structure which was found for both samples. They also should show in a Table the factor loadings for each item for both samples. This would help the reader to get a better idea what items went with what factor.

We agree with the reviewer that accurate tables significantly contribute to clarifying results. Following Journal’s guidelines, we have attached as Supplementary materials all the tables that showed the results of the EFAs solution explored. In these tables, we reported the factor loadings index for each item and the internal consistency indices of each factor. For the SZ group please refer to tables S4-S7, while for the GP group please refer to tables S8-S10.

Were there items which loaded on both factors (SR and SC)?

As suggested by Stevens (1992) we adopted a factor loading index of ≥|.40|; all the items lower than this threshold were dropped. Moreover, the best practice in exploratory factor analysis suggests dropping the cross-loading items. For all the details, please refer to the tables in Supplementary Materials.

The GP sample lost five items from the original subscale. Again, one would want to see what the factor loadings of these items were and which exact items.

We agree with the Reviewer that a better understanding of the statistical process is facilitated by detailed information. Thus, we reported all the necessary information in the supplementary materials.

Discussion:

They showed that their analysis in the SZ supported a 2-factor model, similar, but not equal to Beck’s original structure of the scale. Could they speculate more whether the Italian language played a role in this difference?

The issue highlighted by the Reviewer is quite interesting, and it is an argument that initially we have considered. However, it is hard to demonstrate that the Italian language (or any other language) can be responsible per se for a different factor structure. Considering the literature, and thus the evidence drawn from other validation studies of the BCIS, we are more inclined to support the hypothesis that our results are mainly due to the characteristics of our samples. As we have pointed out in the Introduction and the Discussion sections, the depletion of items of the BCIS is not so uncommon, as the outcome of previous studies and at the same time a fair heterogeneity in indices and factor structure. Anyway, the point raised by the Reviewer deserves attention, thus we have stressed the issue in the Conclusions section widening this point as follows: “In particular, the depletion of some items from the original form of the scale cannot be disregarded. We may wonder what factors led to this evidence. It might be traced back to the translation into the Italian language, but we are not inclined to consider this hypothesis, as the translation process was supported by good results. Furthermore, the suitability of eliminating some items from the BCIS even in the English language, is not new at all. The appropriateness of some items [13,50,54] is controversial also in previous studies, raising doubts about the suitability of some items in healthy subjects, saving items not reaching the loading threshold to guarantee consistency with Beck’s original results, or even suggesting re-naming the original dimensions [13,17,20,25,50].” (lines 490-499)

In the limitations, they need to indicate that they did not examine reliability, test-test stability and discriminant validity

According to the Reviewer’s suggestion, we have included the following statement in the Limitations paragraph: “Fourth, our study aimed at exploring specifically the underlying factor structure of the Italian version of the BCIS, but did not include the investigation of additional psychometric properties, such as test-retest stability, and discriminant validity. Further studies to deepen these features are required to provide a fully exhaustive perspective on the issue” (lines 467-471).

Comments on the Quality of English Language

A few English edits are needed.

We have performed a final English editing process.

Reviewer 2 Report

ID: ijerph-2451018

Title: The Italian validation of the BCIS: different evidence in psychotic patients and the general population.

Thank you for providing a chance to review this manuscript.

Detailed information:

Stronger evidence or a more scientific approach is needed to support item deletion. Specific comments are provided below:

       1. Don’t have abbreviations in the title.

2. As an observational study, please follow the STROBE guidelines.

3. The source, composition, inclusion and exclusion criteria, and number of exclusions of the general population are confusing.

4. There is a large gap between the sample sizes of patient and the general population. Are these sample sizes sufficient to support the conclusions of this study?

5. What are the psychometric properties of this scale in other cohorts and in other linguistic contexts? A systematic literature review on this scale is warranted. Since this scale has been translated into so many versions, are there more previous studies that share this research?

6. The scale has different methods of item deletion as well as factor attribution in different cohorts, which largely affects the generalized application of the scale. Based on the results of this study, it is speculated whether the results would be different again if applied in other specific cohorts? And then different methods of item deletion would need to be made? I am skeptical of the feasibility of the scale reduction based solely on the results of this study.

7. Please follow the COSMIN guidelines for psychometric assessments.

8. The way of scale reduction in this study is unconvincing.

Thank you and my best,

Your reviewer

Moderate editing of English language required

Author Response

We sincerely thank the Reviewer for his/her comments that helped us to significantly improve our paper. here below our answers to his/her notes:

Detailed information:

Stronger evidence or a more scientific approach is needed to support item deletion. Specific comments are provided below:

  1. Don’t have abbreviations in the title.

According to the Reviewer’s suggestion, in the title we have replaced “BCIS” with “Beck Cognitive Insight Scale”

  1. As an observational study, please follow the STROBE guidelines.

Consistently with the Reviewer’s indication, we have checked the correspondence of our paper with the STROBE guidelines. Most of our paper was already in line with STROBE recommendations, however, where necessary, we have included and adjusted further details and pieces of information.

  1. The source, composition, inclusion and exclusion criteria, and number of exclusions of the general population are confusing.

We thank the Reviewer for encouraging us to improve this point. Consistently we have slightly rephrased the paragraph to make it clearer (lines 122-134)

  1. There is a large gap between the sample sizes of patient and the general population. Are these sample sizes sufficient to support the conclusions of this study?

We thank the reviewer for highlighting this point. Different researchers questioned the adequate sample size to support EFA results. Just to cite some, Zeller (2006) concluded that a sample between 10 to 50 was sufficient for two dimensions and 20 variables. Other authors (Geweke & Singleton, 1980; Bearden, Sharma, & Teel, 1982) stated that 25-30 subjects can be adequate. Others considered at least 100 subjects as a minimum sample size for consistent factor recovery. Nonetheless, the size of our sample is adequate to support EFA results. As discussed previously by different authors even smaller samples are deemed adequate (Zeller, 2006; Geweke & Singleton, 1980; Bearden, Sharma, & Teel, 1982), and as recommended by the COSMIN guidelines, a sample equal or greater than 100 subjects is the golden standard. We have included this point in the text (line 197).

  1. What are the psychometric properties of this scale in other cohorts and in other linguistic contexts?

We agree with the Reviewer that a concise but exhaustive overview would be appropriate. Thus, in the Introduction and in the Discussion sections, we have included additional paragraph where previous studies are debated more in detail (lines 66-91; 384-399; 424-440)

A systematic literature review on this scale is warranted. Since this scale has been translated into so many versions, are there more previous studies that share this research?

Please see the previous point

  1. The scale has different methods of item deletion as well as factor attribution in different cohorts, which largely affects the generalized application of the scale.

If we have correctly understood the point raised by the Reviewer, the statistical procedure adopted for both groups (SZ and GP) is the same. We performed the following steps:

  • a PCA to extract the factors with eigenvalues greater than or equal to 1.0;
  • an EFA to test the factor solution derived from PCA;
  • rejection of the factor solutions with at least one factor composed only of 2 items, as the EFA best practice suggests dropping these factors since 2 items define only a single correlation;
  • alternative EFAs until the best factor solution in terms of items for each factor and internal consistency is obtained.

In SZ the scree plot reported 6 factors with eigenvalues greater than or equal to 1.0, while in GP 4 factors were found. Therefore, the EFA started from a different factor solution for each group examined; however, the results suggested that a 2 factors solution was the best model for both, considering the related reliability values.

Based on the results of this study, it is speculated whether the results would be different again if applied in other specific cohorts? And then different methods of item deletion would need to be made? I am skeptical of the feasibility of the scale reduction based solely on the results of this study.

The Reviewer raises a very interesting point. Indeed, evidence from our study highlights a different factor structure in different groups. This is not the first case in literature, as we have highlighted in the Introduction and Discussion sections. Previous studies and authors have debated this point stressing the opportunity to include or delete some items in different groups, as well as some validation studies have pointed out a different factor structure. However, we hypothesize these differences in results may be due mainly to not fully clear inclusion/exclusion criteria and to inconsistencies in methodological approach. Actually, the aim of our study was not to cast shades on the BCIS but rather to deepen its purposefulness and to enhance its ability to catch metacognitive attitudes, going beyond the mere role as a complementary factor of insight into illness. As a matter of fact, our study provides new information, and if no methodological errors can blur their validity, they deserve attention and further deepening. Finally, in the Discussion section, we have stressed that additional evidence is required to clear up some points, in the meantime, it is reasonable to adopt the classical formulation of the BCIS.

  1. Please follow the COSMIN guidelines for psychometric assessments.

We thank the reviewer for the suggestion. We have checked the congruence of our text with the COSMIN recommendations. Most of our paper was already in line with them, however, where necessary, we have included and adjusted further details and pieces of information.

  1. The way of scale reduction in this study is unconvincing.

We hope that all the changes, adjustments, and additional information provided in this revised version would make the study and the evidence that emerged more convincing and clearer. We have stressed the coherence and the correctness of the methodological approach adopted, especially regarding the strict inclusion criteria, and above all the high specificity of the diagnostic criteria for the SZ group, as well as the statistical procedures. As stressed elsewhere, the aim of our study was not to cast shades on the BCIS but rather to deepen its purposefulness and enhance its ability to catch metacognitive attitudes, going beyond the mere role as a complementary factor of insight into illness. As a matter of fact, our study provides new information, and if no methodological errors can blur their validity, they deserve attention and further deepening. Finally, in the Discussion section, we have stressed that additional evidence is required to clear up some points, in the meantime, it is reasonable to adopt the classical formulation of the BCIS. We have stressed this point in the Conclusion section as follows: “Nonetheless, the evidence highlighted in our study poses a pragmatic problem for research. We may wonder whether two different forms of the same questionnaires can be administered in a unique study to assess the same construct in GP and SZ patients. In particular, the depletion of some items from the original form of the scale cannot be disregarded. We may wonder what factors led to this evidence. We are not inclined to consider the hypothesis about the translation into the Italian language, as the translation process was supported by good results. Moreover, the suitability of eliminating some items from the BCIS even in the English language, is not new. The appropriateness of some items [13,31,58] is controversial also in previous studies, raising doubts about the suitability of some items in healthy subjects, saving items not reaching the loading threshold to guarantee consistency with Beck’s original results, or even suggesting renaming the original dimensions [13,17,20,25,31]. Notably, we recommend not to interpret the evidence that emerged from our study as questioning the purposefulness of the BCIS per se. No doubt that the informative power of the BCIS showed to go beyond the mere role of a complementary factor of insight into illness, but rather investigates upper metacognitive attitudes. Rather, the partly controversial debate of the underlying factor structure of the scale, with the contribution of our additional data, would require further deepening and clarification in future studies. Waiting for further definitive evidence, we suggest to administer the classical form of the BCIS” (lines 488-506).

Thank you and my best,

Your reviewer

Comments on the Quality of English Language

Moderate editing of English language required

We have performed a further English revision

Round 2

Reviewer 1 Report

The Introduction/Background needs rework:

1.There is a repetitive statement which needs to be eliminated and also referenced in the following passage:

“The BCIS has been widely administered also to healthy subjects. So far, a similar factor 63 structure of the BCIS in the GP and psychiatric patients was found and the two-factor 64 structure has been confirmed in both of the populations. 65 Notably, previous studies highlighted a similar factor structure of the BCIS in GP and 66 psychiatric patients, mostly supporting the original two-factor structure.”

3. The following sentence is incomplete: “Despite this, a 67 discrete variability in reliability results, statistical procedures, and pragmatical considerations [30].”

4. The entire section on discussion of the BCIS between different version/item changes for the general population needs to be placed into the discussion section.

Results:

The two samples are now adequately described. One error in the Table needs to be corrected:

PANSS total cannot be “Min 19”. The lowest PANSS is 30. The title of “Reliability” needs to be changed to Internal Reliability.

Discussion and Conclusions:

There are many unnecessary repetitions and unrelated statements, which need to be deleted. In general, the manuscript should be edited by a native English speaker.

The manuscript needs to be edited by a native English speaker.

Author Response

We sincerely thank the Reviewer for his active support and accuracy which actually helped us to improve our paper. 

Below our point-by-point answers:

1.There is a repetitive statement which needs to be eliminated and also referenced in the following passage:

“The BCIS has been widely administered also to healthy subjects. So far, a similar factor 63 structure of the BCIS in the GP and psychiatric patients was found and the two-factor 64 structure has been confirmed in both of the populations. 65 Notably, previous studies highlighted a similar factor structure of the BCIS in GP and 66 psychiatric patients, mostly supporting the original two-factor structure.”

We thank the Reviewer for pointing out this inaccuracy. We have deleted the repetition and included proper references (lines 63-67)

  1. The following sentence is incomplete: “Despite this, a 67 discrete variability in reliability results, statistical procedures, and pragmatical considerations [30].”

Following the Reviewer’s remark, we have corrected the sentence (lines 67-69)

  1. The entire section on discussion of the BCIS between different version/item changes for the general population needs to be placed into the discussion section.

Following the Reviewer’s suggestion, we have split this paragraph into two parts, that have been eventually included in the Discussion section: one about the SZ group results (lines 401-409) and one about the GP group results (lines 451-468). We have consistently modified the sentence in the Introduction section (lines 67-71).

Results:

The two samples are now adequately described. One error in the Table needs to be corrected:

PANSS total cannot be “Min 19”. The lowest PANSS is 30. The title of “Reliability” needs to be changed to Internal Reliability.

We thank the Reviewer note. Actually, the scores did not refer to the total score of the PANSS, but to the General Psychopathology scale. We have corrected the label in the Table and the legend. To distinguish the acronym “GP” for General Population, exceptionally we have indicated General Psychopathology as “GPsy”

Discussion and Conclusions:

There are many unnecessary repetitions and unrelated statements, which need to be deleted. In general, the manuscript should be edited by a native English speaker.

We agree with the Reviewer that the Discussions section, after all the changes, had to be re-edited. Thus, we have peformed a full revision of the text.

Reviewer 2 Report

ID: ijerph-2451018

Title: The Italian validation of the BCIS: different evidence in psychotic patients and the general population.

Thank you for providing a chance to review this manuscript.

Comment: Accept.

Detailed information:

Thanks to the author for the revisions and explanations. This revision basically addresses my comments.

Thank you and my best,

Your reviewer

Minor editing of English language required

Author Response

We thank the Reviewer for his active support and we are glad that the revised version is satisfactory.

Due to all the progressive changes, we ha ve performed a full English revision of the text.